# Adenosine Targeting as a New Strategy to Decrease Glioblastoma Aggressiveness

**DOI:** 10.3390/cancers14164032

**Published:** 2022-08-20

**Authors:** Valentina Bova, Alessia Filippone, Giovanna Casili, Marika Lanza, Michela Campolo, Anna Paola Capra, Alberto Repici, Lelio Crupi, Gianmarco Motta, Cristina Colarossi, Giulia Chisari, Salvatore Cuzzocrea, Emanuela Esposito, Irene Paterniti

**Affiliations:** 1Department of Chemical, Biological, Pharmaceutical and Environmental Sciences, University of Messina, Viale Ferdinando Stagno D’Alcontres, 31-98166 Messina, Italy; 2Istituto Oncologico del Mediterraneo, Via Penninazzo 7, 95029 Viagrande, Italy

**Keywords:** glioblastoma, adenosine, tumor microenvironment, A2AAR antagonist, immune evasion, adenosine receptors

## Abstract

**Simple Summary:**

Given the rising mortality rate caused by GBM, current therapies do not appear to be effective in counteracting tumor progression. The role of adenosine and its interaction with specific receptor subtypes in various physiological functions has been studied for years. Only recently, adenosine has been defined as a tumor-protective target because of its accumulation in the tumor microenvironment. Current knowledge of the adenosine pathway and its involvement in brain tumors would support research in the development of adenosine receptor antagonists that could represent alternative treatments for glioblastoma, used either alone and/or in combination with chemotherapy, immunotherapy, or both.

**Abstract:**

Glioblastoma is the most commonly malignant and aggressive brain tumor, with a high mortality rate. The role of the purine nucleotide adenosine and its interaction with its four subtypes receptors coupled to the different G proteins, A1, A2A, A2B, and A3, and its different physiological functions in different systems and organs, depending on the active receptor subtype, has been studied for years. Recently, several works have defined extracellular adenosine as a tumoral protector because of its accumulation in the tumor microenvironment. Its presence is due to both the interaction with the A2A receptor subtype and the increase in CD39 and CD73 gene expression induced by the hypoxic state. This fact has fueled preclinical and clinical research into the development of efficacious molecules acting on the adenosine pathway and blocking its accumulation. Given the success of anti-cancer immunotherapy, the new strategy is to develop selective A2A receptor antagonists that could competitively inhibit binding to its endogenous ligand, making them reliable candidates for the therapeutic management of brain tumors. Here, we focused on the efficacy of adenosine receptor antagonists and their enhancement in anti-cancer immunotherapy.

## 1. Introduction

Brain tumors affect different areas of the brain, including the cerebellum, and portions of the central nervous system (CNS), such as the spinal cord, that normally control voluntary and involuntary functions [1]. Physiologically and anatomically, the brain is separated from the blood by the blood–brain barrier (BBB), made up of tightly junctioned endothelial cells, astrocytes, and pericytes, which selectively control the exchange of substances between the two compartments [2]. Being intracranial, tumor cell growth causes an increase in the tumor mass that compresses the blood vessels, initiating a tumor-associated cerebral edema process that compromises the integrity of the BBB itself, and causes an active outflow of molecules [3]. It has been reported that the tumor mass increases the intracranial pressure that disrupts the homeostasis of the brain-affected area surrounding the tumor site and generates secondary effects [4]. 

Although clinical symptoms depend on the location and size of the tumor mass, the most common symptoms of all brain tumors include headache, seizures, nausea and vomiting, confusion and disorientation, loss of balance or dizziness, and memory loss (Figure 1B) [5]. The specific causes of brain tumors are unclear, but a set of risk factors should be considered in the tumor development and progression: age, exposure to ionizing radiation, decrease in immune defenses, and genetic predisposition of proto-oncogenes [6].

Still, CNS tumors are rare; they represent 1.6% of all tumors, especially in the last three decades, during which a progressive increase in the incidence of brain tumors development has been reported, particularly in adults over 65 years of age [7].

Brain tumors are classified into primary tumors, which originate in the brain, and can be benign and malignant tumors, and secondary tumors, which represent metastatic tumors [8]. A second classification is based on the origin of the cells from which brain tumors derive, such as glial cells, responsible for myelin production and represent 40% of all primary tumors and 78% of all malignant tumors [9]. One of the most common gliomas is astrocytoma, which originates from astrocyte cells and is distinguished by pilocytic tumors, very rare tumors that account for about 5% of gliomas and are more common in children [10]. Unlike other gliomas, these rarely develop into more aggressive tumors and can be cured with surgery [11]. There are also diffuse and anaplastic astrocytomas consisting mainly of immature cells, which, over time, tend to transform into more aggressive forms [12]. Oligodendroglioma originates from oligodendroglia cells and includes tumors with different grades and aggressivity, and ependymoma originates from ependymal cells that line the ventricular system and represent 2–3% of all brain tumors (Figure 1A) [13]. Finally, glioblastoma multiforme (GBM) is the most common, invasive, and malignant of all brain tumors, characterized by rapid growth and the invasion of adjacent regions of the brain, such as the meninges or cerebrospinal fluid [14]. 

According to the World Health Organization (WHO), GBM accounts for 75% of aggressive malignant brain tumors in adults [15] and has been classified as a grade IV tumor. Moreover, a further classification divides GBM into a primary tumor [16] that is aggressive and invasive whose are not derived from other diseases, and which represents the most common form (about 95%), affecting the elderly population. It is also classified as a secondary tumor [16], which derives instead from low-grade astrocytomas, with an incidence rate in young people [17]. Human GBM is a heterogeneous tumor consisting of tumor cells and a small portion of cancer stem cells (CSCs), which have a high tumorigenic potential and are characterized by excessive proliferation, invasiveness, and metastasis [18], and by high resistance to standard therapy, in which case the patient’s estimated time of survival is about 15 months from the first diagnosis [19].

It was initially thought that GBM derived exclusively from glial cells, but scientific evidence has shown that there are other native cell types, similar to neuronal stem cells, from which GBM originates [20]. Glia, neurons, and stem cells show alterations in their functions that contribute to the development and progression of the tumor. There are two forms of heterogeneity in GBM: the intertumor and the intratumor. First, intertumoral heterogeneity refers to the genetic alterations that occur in individual tumors [21]. Secondly, intratumoral heterogeneity refers to the diversity of the phenotypes of the tumor cells that make up the tumor mass and other cellular entities recruited into TME, such as microglia and/or macrophages and endothelial cells [22]. However, it is difficult to identify their cell of origin [23] and remains a subject of debate to this day. Initially, considering the ability of astrocytes to replicate in the adult brain, it was hypothesized that an oncogenic alteration of astrocytes could give rise to GBM [23]; later, other hypotheses were developed, such as derivation from precursors such as oligodendrocytic or neuronal cells [24]. Recent studies have focused on glioma stem cells, which have the ability to self-renew and form tumors in vivo with the same characteristics as the primary tumor [25], offering greater resistance to chemotherapy, thus associating their presence with tumor heterogeneity [25,26]. Moreover, various pathways and molecular mechanisms are involved in the development and progression of GBM [27], including the microenvironment of GBM, characterized by hypoxic regions [18] whose oxygen deficiency determines the activation of the transcription factor; the hypoxia-inducible factor (HIF) [28], inducing gene transcription that promotes the production of various pro-angiogenic factors including vascular endothelial growth factor (VEGF) [29], which leads to the formation of new blood vessels; fibroblast growth factor (FGF) [30], which promotes the proliferation and differentiation of endothelial, smooth muscle, and fibroblast cells; and finally platelet growth factor (PDGF) [31], released by platelets and stimulating the proliferation and migration cancer cells, all factors that promote the progression of GBM.

The immune system is also an essential component for tumor development and progression, especially in GBM [32]. Activation of the immune system contributes to creating an immunosuppressive microenvironment in GBM [33], which consists of the production and release of immunosuppressive cytokines and chemokines, such as transforming growth factor-β (TGF-β), interleukin 10 (IL-10), prostaglandin E2 (PGE2), and many immune cells, such as immunosuppressive natural killer (NKT) cells, regulatory T/B cells (T/B-reg), tumor-associated macrophages (TAMs), and myeloid-derived suppressor cells (MDSC) [34,35].

The hypoxic state of the GBM microenvironment results in an increase in the concentration of adenosine, a small nucleoside and key mediator of several biological functions, thus defining it as tumor-derived adenosine [36]. In fact, adenosine through interaction with its four G protein-coupled receptors (GPCRs), A1, A2A, A2B, and A3, is involved in the blocking of anti-tumor immunity. Confirming what has been written, Hoskin et al. showed that adenosine, present in the GBM microenvironment, can inhibit the function of natural killer cells, as well as the ability of cytotoxic T cells to adhere to tumor cell targets [36]. In particular, it is thought that it is GBM itself, through activation of the A2A receptor, above all, which causes adenosine to carry out an activity opposite to the biological one. In fact, in this regard, almost 20% of human cancers contain mutations in genes that code for GPCRs [37]. By exploring the sequencing data from the Genomic Data Commons data portal (GDC), cancer-associated mutations of the A2A receptor affecting its activity were identified, further confirming the involvement of adenosine and its receptor in GBM development [38].

Treatment of GBM requires a multidisciplinary approach and consists of surgical excision, followed by radiation treatment with concomitant temozolomide (TMZ), and finally chemotherapy with TMZ [39], although, an unfavorable condition of combined treatment of radiotherapy and high-dose TMZ is lymphopenia, defined as a reduction in peripheral blood lymphocytes [40]. Where surgical excision is not possible, other therapeutic approaches are used, some of which have already been approved by regulatory agencies in the United States and consist of the use of monoclonal antibodies (mAb), such as Bevacizumab [41], a humanized mAb that blocks VEGF, blocking angiogenesis. Regorafenib is involved in GBM relapse, which acts on the processes of angiogenesis, with the modification of TME [42]. Tumor treatment field therapy (TTF) is a non-invasive anti-cancer treatment [43] that uses alternating electric fields tuned to low intensity (1–3V/cm) and intermediate frequency (100–300 kHz) to stop the splitting of solid tumor cells. This therapy has limitations due to its mechanism of action, which blocks biological processes, such as DNA repair mechanisms, and its cost [43,44]. Finally, gene therapy represents the recourse in the therapeutic approach for the treatment of GBM, which consists of the use of genetic fragments in association with viral vectors [45,46] capable of replicating within tumor cells, causing their death and also releasing viral particles capable of infecting and killing adjacent cancer cells [47]. In the context of GBM, considering the presence of immune cells with immunosuppressive action, an immune checkpoint has recently been identified, represented by BACE1; although BACE1 is a beta-secretase involved in the cleavage of the amyloid precursor protein [48] and used in the treatment of Alzheimer’s disease (AD), its inhibition was effective for GBM. This approach represents an opportunity to safeguard depleted T lymphocytes, while also inducing increased infiltration of CD8^+^ T lymphocytes into the GBM [49]. In addition, the use of chimeric antigen receptor T cells (CAT-T), which produce T cells capable of acting in TME, has been shown to be effective. Several antigens have been targeted with this technique, optimizing the anti-tumor activity [50]. A last therapeutic strategy recently discovered and still under development concerns the use of adenosine receptor antagonists [51], which have been shown to prevent the effects related to extracellular adenosine.

## 2. Adenosine and Adenosine Receptors (ARs)

Adenosine is a small molecule present throughout the human body, capable of performing various physiological functions following interaction with its receptor subtypes (A1, A2A, A2B, and A3). It is present in the cardiovascular system, where it modulates the vasoconstriction and vasodilation of arteries and veins [52]; in the metabolic context, adenosine inhibits lipolysis and induces bronchoconstriction [53,54], and regulates diuresis, muscle tone, and locomotion. At the level of the CNS, it exerts neuroprotective activity against ischemic events [55], hypoxia, and oxidative stress, and modulates the release of neurotransmitters; it is also involved in the regulation of cytokines and the production of T lymphocytes by the immune system [56,57]. There are two forms of adenosine: intracellular and extracellular [58], widely expressed in all tissues, and obtained by the dephosphorylation of its precursors, adenosine triphosphate (ATP), adenosine diphosphate (ADP), and adenosine monophosphate (AMP), or by hydrolysis of S-adenosylhomocysteine (SAH) [59]. Physiologically, the intracellular concentration of adenosine is regulated by an important enzyme known as adenosine kinase (ADK), and by two transporters: the equilibrative nucleoside transporters (ENT) and the bidirectional passive transporters, which play a critical role, as they allow free movement of adenosine across the cell membrane [60], and nucleoside concentrative transporters (CNTs), Na-dependent transporters that coordinate the adenosine gradient transport [61]. The direction of this nucleoside, absorbed or released by cells, is determined by the difference in concentration between the two forms, intracellular and extracellular, across the membrane [58]. Adenosine is defined as a “helper” in protecting cells such as neurons and cardiomyocytes against stressful conditions, allowing them to regulate their activity to reduce ATP requirements and ensure cell survival [58]. This is possible because adenosine can be released into the extracellular environment, where it acts as a specific modulator through cell surface receptors [56]; these receptors, called “ado receptors” (Ars), are GPCRs and are classified into four subtypes: A1, A2A, A2B, and A3 [62], which differ in the number of amino acids (Figure 2) and in their affinity towards adenosine. In fact, A1 and A2A possess a high affinity for adenosine compared to A2B and A3, which have a low affinity for the nucleoside [63].

Among them, A1A and A3A receptors are coupled to Gi and Go proteins that inhibit adenylate cyclase activity and reduce intracellular cAMP levels. This will result in the activation of phospholipase C(PLC)-β, thereby increasing inositol-1,4,5-triphosphate (IP_3_) [64] and intracellular calcium levels, which in turn stimulate activation of the Ca-dependent protein kinase (PKC) and all calcium-binding proteins [64]. In the CNS, A1AR is expressed in microglia/macrophages and neurons, and plays a crucial role in their activation [65]; peripherally, it is also highly expressed in cardiac, renal, and adipose tissue. As demonstrated by Synowitz M. et al. in A1AR knockout mice, there is an increase in neuroinflammation and microglia activity [66], and this suggests that, in pathological conditions, A1AR activation produces a neuroprotective effect [67]. In physiological conditions, adenosine, through A1AR, determined a decrease in the proliferation of astrocytes, inducing the release of neurotrophic growth factor (NGF) [68]. A3AR, however, has a low expression in the CNS, but it is highly expressed in immune cells [59], cardiac cells, epithelial cells, colon mucosa, lung parenchyma, and bronchi. It is demonstrated that A3AR is expressed in cells involved in inflammatory processes, suggesting its potential involvement in inflammatory pathologies, such as lung injury, autoimmune diseases, and eye diseases [69]. Moreover, A3AR is present in many types of tumor cells, including astrocytomas, lymphoma, GBM, and other types of cancers [70].

A2AA and A2BA receptors are coupled to the Gs proteins, activating adenylate cyclase and increasing intracellular cAMP levels [71]; moreover, A2AAR activation can promote Protein kinase C (PKC) activation into cyclic AMP-dependent or independent mechanisms [72]. A2BAR activation, however, can stimulate PKC activity by coupling with the Gq protein [73]. They are mainly expressed in the CNS, especially in pre-synaptic regions of the hippocampus, where the release of neurotransmitters such as glutamate, acetylcholine, GABA, and noradrenaline is modulated [74,75], and in post-synaptic regions of the basal ganglia, where they modulate neuronal plasticity. They are also expressed in the astrocytes and oligodendrocytes [76,77] and on the cell surfaces of the immune system [78], such as regulatory T cells, macrophages, and natural killer cells (NKCs) [79], suggesting that they could be valid candidates for cancer immunotherapy. All subtypes of adenosine receptors are expressed on the surface of immune cells, such as macrophages and monocytes, and their expression is regulated by pro-inflammatory cytokines, especially IL-1B and the tumor necrosis factor (TNF) [80], which determined an increase in A2AAR levels on human monocytes [81]. The same pro-inflammatory stimuli regulate the expression of the A2BAR of the macrophages [82]. In physiological conditions, central A2AAR increases NGF and brain-derived neurotrophic factor (BDNF) levels from the hippocampus and cortical neurons [83]. Therefore, given both the prevalence of A2AAR in the CNS and its expression regulated by pro-inflammatory cytokines, this receptor plays a crucial role in inflammatory processes involving microglia, determining the release of IL-1β and IL-18 [84]. In fact, an antagonistic action against A2AAR prevents hippocampal neuroinflammation and IL-1β-induced exacerbation of neuronal toxicity [85]. Evidence showed that, in spinal intermediate neurons of the striatum, this receptor is related to the dopaminergic D2 receptor, where direct and indirect interactions with cholinergic, GABAergic, dopaminergic, and glutamatergic systems have been described, both in the basal ganglia and in other brain structures [86]. In the periphery, A2AAR is localized in the vascular smooth muscle and, together with A1AR, exerts a vasodilatory action. In this context, at the coronary levels, vasodilation mediated by the activation of A2AAR and A1AR is induced by the endothelial enzyme nitric oxidase synthase [87], producing large quantities of nitric oxide and inducing an increase in coronary flow, thus exerting a cardio-protective role [87], and this action depends on an increase in the intracellular cAMP levels [87]. Depending on the location of its receptors by which it interacts, adenosine exerts multiple physiological actions, including the protection of normal tissues and organs from the autoimmune response of immune cells, following binding with A2AAR [88]. In this regard, following damage to the cell, such as hypoxia, inflammation, and tissue injury, ADK activity is reduced, leading to increased levels of extracellular adenosine [59] in the extracellular space, which modulates the immune response, thus containing the inflammatory damage tissue. However, chronic exposure to extracellular adenosine can be harmful in some conditions, as adenosine itself can create an immunosuppressed niche, which is necessary for the development of neoplasia and infections [89,90]. In this context, it has been observed that T_reg_ cells can release ATP, convert it to adenosine, and cause cytotoxic T cell suppression in the local tumor environment [91].

## 3. The Role of Adenosine in Glioblastoma Multiforme

Studies report that extracellular adenosine is an important regulator of several aspects of tumorigenesis, angiogenesis, tumor cell growth, and metastasis [92]. Kezemi et al. provide an interesting review of the expression of adenosine receptors in different tumor cell lines and their effect, including proliferative and tumor-protective expressions, following their activation [93].

It is hypothesized that in the brain, ATP released from the pre- and post-synaptic terminals of neurons and glial cells is the source of extracellular adenosine [94]. In the extracellular area, adenosine is produced from ATP after dephosphorylation by specific ectoenzymes, in this case, CD39 and CD73, expressed in microglial cells [90]. In physiological conditions, CD39 and CD73 exert an important role in the purinergic signals delivered to immune cells through the conversion of ADP/ATP to AMP to adenosine [95]. The CD39/CD73 pathway changes with the pathophysiological context in which it is embedded [89]. It has been demonstrated in vivo study that mice deprived of CD73 presented a lower level of extracellular adenosine, suggesting that ATP degradation is the main source of extracellular adenosine [96]. CD39 is expressed on the surface of the regulatory T cells, and it is the dominant ectoenzyme that controls extracellular nucleoside concentration [90]. Considering that angiogenesis is an important process for the growth of the tumor cells, it has been demonstrated that in mice deprived of CD39, angiogenesis is blocked, causing a slowdown in tumor growth [95].

High concentrations of adenosine and its receptors have also been found in the interstitial fluid tumor, modulating tumor growth [97]. Since the TME contains high levels of extracellular adenosine, it is hypothesized that tumor-derived adenosine is a mechanism by which tumors evade the immune response [98,99]. This evasion strategy is due not so much to the inability of immune cells to recognize the tumor, but the failure of the immune system to activate in the presence of the antigen [100] due to the inhibition of T cells by adenosine itself [100].

It is known that the immune system, through antigen-presenting cells (APC), is able to recognize a specific antigen [101], subsequently allowing the binding with B and T lymphocytes through their receptors, B-cell receptor (BCR) and T-cell receptor (TCR), respectively, thus initiating the immune response. In this tumor context, the activation of the immune system will lead to the secretion of anti-cancer cytokines [102], such as Interferon-gamma (INF-γ), TNF-α, and IL-6, and cell phagocytosis to eliminate the tumor, thus becoming a tool for the development of new treatments in cancer therapy [103]. Nevertheless, most tumors are able to implement various mechanisms to evade the immune response, such as inhibiting tumor-specific immune cells [104]. As is often the case, a particular tumor may express an antigen that, if presented by resting cells or by unprofessional APCs, recognition of the TCR will not lead to tumor destruction, but to inactivation of the tumor-specific T cell [105] (Figure 3).

An important aspect of the mechanism of escape by the tumor from the immune system is the TME [106], which is characterized by a hypoxic state and is rich in inhibitory ligands and cytokines, such as IL-10 and TGF-β, which lead to tolerance by the immune cells towards the tumor [107,108] (Figure 3). These conditions determine the increase in the expression of CD39 and CD73 [89], present on the surface of the tumor by stimulating the production of extracellular adenosine, by activating A2AAR. Moreover, at the same time, there is a reduction in the activity of the adenosine metabolizing enzyme, ADK [109].

In addition, it has been reported that the deletion of functional adenosine receptors, in particular A1AR, results in increased GBM growth [66]. However, subsequent studies have found that the interaction of adenosine with A2AAR induces inhibition of the adaptive immune response, inhibiting the function of CD4^+^ and CD8^+^ T cells and NKCs and IL-2/Nkp46-activated NK cells specifically via A2AAR [110], thus promoting tumor escape from the immune system and metastasis [111,112]. Several in vitro and in vivo studies report that genetic deletion of the A2AAR enhances the anti-tumor responses, confirming adenosine’s role in evading the tumor from the immune system [113]. Second, adenosine appears to block both the generation and effector phases of anti-tumor responses. In vitro studies have been conducted on GBM cell lines U87MG, U373MG [59], and ASB19, which were subjected to hypoxia [59] for 24 and 72 hrs using ATB702 dichloride hydrate (15uM), an ADK inhibitor, and resulted in an accumulation of adenosine [114]. Subsequently, the cells were treated with TMZ (100 µM), which resulted in a decrease in the vitality of the tumor cells compared with the control GBM cells, thus demonstrating the tumor-protective role of endogenous adenosine against TMZ [115].

It has been shown that extracellular adenosine, defined as an immunosuppressive factor through interaction with its receptor, exploiting the hypoxic condition of the TME [116], is able to lead to an increase in intracellular cAMP, inhibiting lymphocyte-mediated cytolysis and, consequently, functional inhibition of immune cells, thus acting as a protective shield against the tumor, helping it to evade the immune system [99]. Therefore, if GBM cells contribute to immunosuppression, the immune cells recruited into the tumor may also participate in its immune escape [32]. Indeed, most of the anti-tumor immune cells recruited to the TME adopt an immunosuppressive phenotype due to the cytokines secreted by GBM [32].

In this context, a large number of experiments have shown that the concentration of adenosine in the TME is much higher than in normal tissues [117].

Hypoxia and tissue damage are not the only factors determining the release of extracellular adenosine; it is also generated from extracellular nucleotides by ectonucleotidases [95] CD39 and CD73 [89]. Through clinical studies, CD73, rather than CD39, was found to be a critical component in adenosine accumulation and tumor immunosuppression. Indeed, overexpression of CD73 was reported to be a component of glioma cell adhesion and tumor cell–extracellular matrix interactions [118].

Moreover, high adenosine concentrations also induce receptor-independent reactions by reversing the reaction catalyzed by S-adenosylhomocysteine hydrolase (SAH-hydrolase), leading to an accumulation of SAH-inhibiting methyltransferases [119], as was shown in a recent study in which adenosine induced DNA hypomethylation in the brain by inhibiting trans-methylation reactions [120]. This connection between adenosine and methyl group metabolism is important for diagnostic purposes because an alteration in methyl group metabolism has been shown to be a risk factor in brain diseases such as GBM and neurodegenerative diseases [121,122].

In vitro and in vivo studies have shown that the presence of adenosine receptors in microglia is well established [123]. Cell cultures of rat microglia specifically express the A2AAR and were treated with the specific agonist CGS21680, inducing the expression of K^+^ channels, which are linked to microglia activation [124]. Again, there is conflicting evidence regarding the role of this receptor: stimulation of the A2AAR in rat microglia induces the expression of nerve growth factor and its release, thus exerting a neuroprotective effect [125], and at the same time induces the expression of Cyclooxygenase-2 (COX-2) in rat microglia by releasing prostaglandin [126].

To confirm the involvement of adenosine and its receptors in tumorigenesis and its tumor-protective role, in vivo studies were conducted using the adenosine receptor agonists or antagonists [127]. A2AAR blocking using SCH58261, an A2AAR antagonist, inhibited the tumor growth, reducing CD4^+^ and regulatory T cells, and improving the anti-tumor response by T cells [127].

There is conflicting evidence regarding adenosine-mediated receptor actions on GBM proliferation. In glioblastoma stem cells, activation of A1AR and A2BAR appears to have reduced tumor proliferation and induced apoptosis [119], whereas in non-glioblastoma stem cell lines, activation of A1, A2B and/or A3 receptors induced an increase in proliferation. Liu et al. reported a pro-proliferative action of adenosine mediated by activation of the A2B receptor on glioblastoma cell lines subjected to hypoxia [128].

## 4. Adenosine Receptor Antagonists

The A2R-mediated adenosine pathway and its immunosuppressive role has allowed research to focus on novel therapeutic approaches to provide prolonged life expectancy in patients with tumors refractory to other therapies. One such approach that has proved effective in enhancing immunotherapy involves the development of selective adenosine receptor antagonists [129], which are able to prevent the effects of extracellular adenosine produced by both tissue and T cells [127]. Given that the first clinical trials of A2AAR antagonists date back to 2020, further clinical studies are underway regarding the anti-tumor activity, as well as the efficacy, of these new therapeutic candidates.

Adenosine receptor antagonists belong to a variety of chemical classes and are divided into two groups: xanthine derivates, of which the best known are Istradefylline (KW-6002) [130] and 3,7-dimethyl-1-propargylxanthine derivates (DMPX), and non-xanthine derivates, the Polyheterocyclic nitrogen system, especially Ciforadenant (CPI-444) and Imaradenant (AZD4635) [131] (Figure 4).

### 4.1. Xanthine Derivates

Compounds belonging to this group result from modifications of the two main alkaloids, caffeine and theophylline [132]. These derivates show a high affinity for all the adenosine receptors, but in the GBM context, receptor affinity must be directed towards A2AAR in order to bind it selectively and competitively, reducing adenylate cyclase activity [133]. The main A2AAR xanthine antagonists are 8-(3-chlorostyryl) caffeine (CSC,7), 1,3-dipropyl-7-methyl-8-(3,4,-dimethoxystyryl)xanthine (KF 17837), 3,7-dimethyl-1-propargylxanthine derivates (DMPX), and Istradefylline (KW-6002), which has a K_i_ of 2.2 nM, and is an extremely strong, selective, and orally active adenosine A2A receptor antagonist [134]. DMPX was the first selective A2AAR to be detected [135]. Many selective A2AAR antagonists have been obtained, some of which are being used in clinical trials for neurodegenerative diseases such as Parkinson’s disease, given the interconnection between dopaminergic D2 receptors and adenosine [136,137].

### 4.2. Polyheterocyclic Nitrogen System

Another group of A2AAR antagonists is represented by the polyheterocyclic nitrogen system, including Preladenant (SCH-420814), Ciforadenant (CPI-444) [138], Taminadenant (NIR178) [139], Imaradenant (AZD4635) [131], SCH442416 [140], and ZM241385 [141], characterized by small molecules that selectively bind to the A2A receptor, competitively inhibiting adenosine binding and signaling [142]. In the GBM context, this compound presents troubles that prevent its use in clinical trials, as it has a high binding affinity for the A2B receptor subtype [138]. With a K_i_ of 0.048 for human A2AAR, the SCH442416 antagonist is considered a strong, selective, and brain-penetrant antagonist of A2AAR [143]. However, strong evidence has been shown by Ciforadenant being active in multiple preclinical tumor models, both as monotherapy and in combination with PDL1 targets, and it has over 66-fold selectivity over the adenosine A1 receptor [144] (Table 1). Clinical studies conducted on GBM patients under 1 and 4 months of treatment with Ciforadenant have shown that it possesses immunomodulatory effects [145]. Through in vitro studies, it was possible to characterize adenosine-related gene expression with the production of chemokines and cytokines, including CXCL5, CCl2, IL-8, and CXCL1, of monocytic, CD14+ origin, using the receptor agonist NECA [146], and how Ciforadenant is able to neutralize them [145]. Thus, these reports suggest that adenosine signaling not only directly reduces T lymphocyte immunity but also shifts the balance from T effector responses to both recruitment and myeloid suppressor functions [138]. 

### 4.3. Enhancement of Immunotherapy Induced by Adenosine Receptor Antagonists

Another therapeutic approach to enhance immunotherapy targets the immune cells in the TME [56]. As previously reported, the A2AAR is expressed on the surface of many cells of the immune system, the activation of which induced an immunosuppressive effect [56]. Consequently, a selective A2AAR antagonist reducing intracellular cAMP levels allows lymphocytes to effectively fight tumor cells. Since A2A and A2B adenosine receptors are coupled to the G proteins, and both increase intracellular cAMP levels [71], the use of A2BAR antagonists leads to a reduction in cAMP by restoring the anti-tumor functions of lymphocytes [145]. It is of interest to note the relationship between A2A and A2B adenosine receptors: A2A is involved in the expression of A2BAR [67]; furthermore, its activity is influenced by the expression of A2BAR, and both proteins can interact to form new functional units [147]. This evidence, therefore, suggests that blocking these receptors may be an effective means of combating cancer [147]. In this regard, clinical trials are already underway in patients with different tumor types where either the use of selective antagonists for individual receptors or dual antagonists is employed [56].

For the treatment of advanced malignant tumors, such as GBM, a trial has been updated for 2021, showing the combined treatment of the dual A2AAR/A2BAR antagonist with the AB122 antibody and standard chemotherapy [56].

## 5. Future Perspectives

Given the increasing mortality rate caused by GBM and despite the many strategies adopted to counteract the tumor’s progression, current treatments do not appear to be effective in influencing tumor growth. It is clear that each tumor is unique and different from the others, and that it is why the development of drugs capable of blocking its growth and at the same time reducing the impact of therapy on the body requires a better understanding of the mechanisms triggered by the tumor. Here, the knowledge about the adenosine pathway and its involvement in the TME has been established and would support research and the discovery of new strategies, including the development of adenosine receptor antagonists. Other approaches to boost immunotherapy would be the use of dual antagonists, both for the A2A and A2B adenosine receptors, given their functional interconnection as immunosuppressive agents. 

## 6. Conclusions

Despite the positive results and the numerous clinical trials underway, there are limitations to the development of these new candidates due to the pharmacokinetic complexity. Therefore, considering the promising effect of adenosine receptor antagonists, they could represent alternative treatments for GBM by using them alone or in combination with chemotherapy to improve patients’ quality of life.

## Figures and Tables

**Figure 1 cancers-14-04032-f001:**
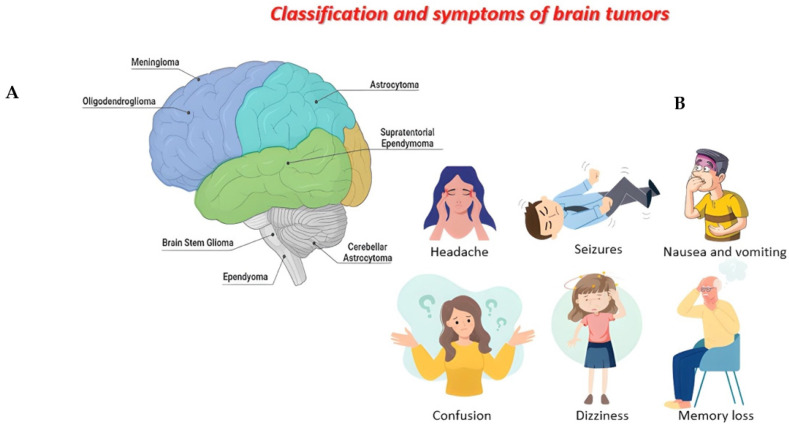
Figure created with Biorender.com. Classification of brain tumors (**A**). Main symptoms of brain tumors (**B**).

**Figure 2 cancers-14-04032-f002:**
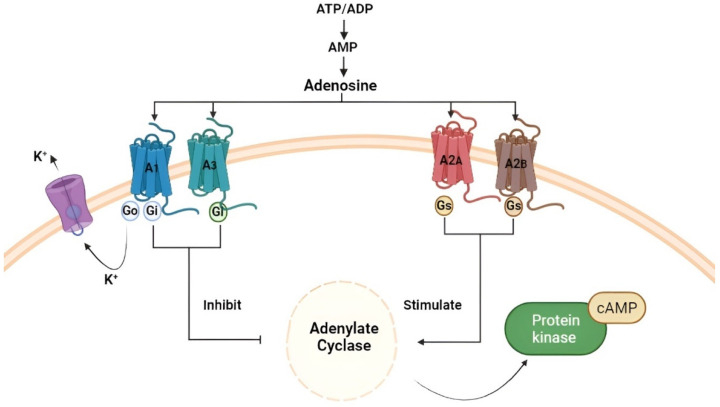
Figure created with Biorender.com. Production of adenosine starts from its precursors ATP, ADP, and AMP and subsequent binding of the nucleoside with the respective receptors A1, A2_A_, A2_B_, and A3.

**Figure 3 cancers-14-04032-f003:**
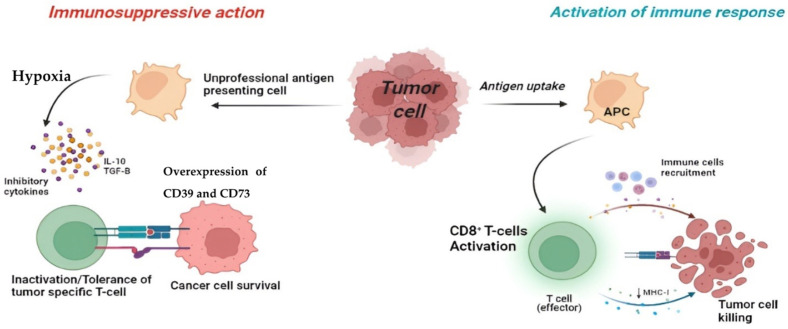
Figure created with Biorender.com. **Right:** induction of the immune response by antigen presenting by APC, with consequent activation of effector CD8^+^ T cells and recruitment of immune cells, with the elimination of the tumor cell. **Left:** lack of immune response, following antigen presentation by non-professional APC cells. The tumor microenvironment, characterized by hypoxia and the presence of inhibitory cytokines and cytokines such as IL-10 and TGF-β, leads to inactivation and/or tolerance of effector T cells with the consequent escape of the tumor from the immune response.

**Figure 4 cancers-14-04032-f004:**
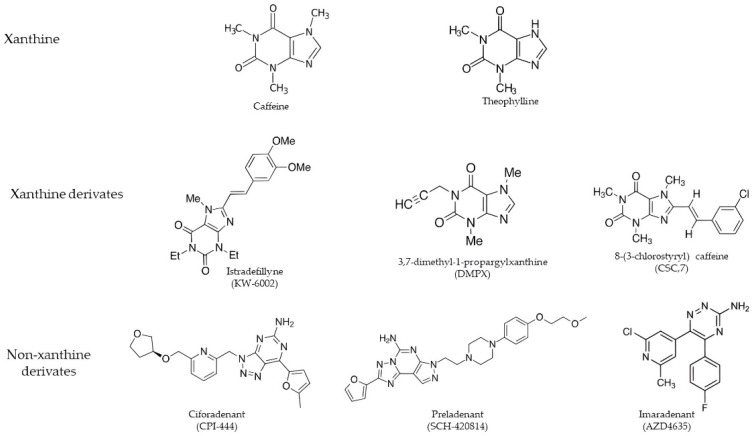
The main adenosine receptor antagonists divided into xanthine derivatives and non-xanthine derivatives.

**Table 1 cancers-14-04032-t001:** Summary of the binding selectivity of adenosine receptor antagonists.

Summary of Adenosine Receptor Antagonists					
Adenosine Receptor Antagonists	A1	A2_A_	A2_B_	A3	References
Xanthine derivates					
Caffeine	+	+	+	+	[132]
Theophylline	+	+	+	+	[132]
DMPX	+	+++	++	+	[135]
Istradefylline	+	+++	+	+	[130]
Taminadenant	+	++	+	++	[139]
Non-Xanthine derivates					
Ciforadenant	+++	+++	++	+	[138]
Imaradenant	++	+++	+	+	[131]
SCH442416	+	+++	++	+	[140]
ZM241385	+	+++	++	++	[141]

+ Low selectivity for adenosine ++ Halfway selectivity for adenosine +++ High selectivity for adenosine.

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
