# Peer review of "Adenosine Targeting as a New Strategy to Decrease Glioblastoma Aggressiveness"

_cancers, 2022, doi:10.3390/cancers14164032_

Round 1

Reviewer 1 Report

This is a short review of the complex field of adenosine receptors, in context of GBMs and brain tumors. The authors succinctly summarize the most important findings that would be of interest to a novice in the field. Necessarily, some of the complexity in ARs, its biological actions, and the challenges/limitations of targeting ARs with respect to GBMs is lost in the process, and could be addressed in revisions to make this review more informative and interesting to the reader.

Some of issues that need to be addressed prior to publication are listed below:

Line 18 – extra period within the sentence pear.to

Line 42 – adenosine binds with its receptors…missing s at the end of bind

The description of points (B) to (F) in the text are mixed up:

Line 45 – (B) and (C) are interchanged/mislabeled in the graphical abstract

Point C which is written as a pathological state…is unclear, when referred to the image that shows blood vessels and its regulation.

Line 54 – The description of the last sub-point (D) in the text should be for point (E) in the image.

Line 56 - The description of the last sub-point (E) in the text should be for point (F) in the image.

Line 100 – Referring to the image creation software as a separate point (B) is confusing because the image contains (A) and (B) which are referred to differently in the preceding text. The same issue also exists in the graphical abstract with point (F). Figures 2 and 3 do not have a separate point referring to the software used.

Page 4, lines 149-166 – As written, this para does a cursory job of introducing adenosine, its complex biological roles, natural origins, differences of its activities compared to ATP/AMP, and receptor subtypes. It would be useful if the expression patterns of different receptors in the body, specially the brain/CNS is discussed. This paragraph should be combined with section 3 (page 6, line 247).

Page 5 – section 2, lines 167-205: the discussion about GBM pathophysiology should be combined with section 1 (introduction) and should come before discussing adenosine and its receptors.

Page 5 – Section 2.2, lines 221-246 (page 6) should be combined with the discussion of current treatment approaches in the introduction section on page 4.

Page 6, line 247 – Section numbering should be 3, not 2.

Page 8, line 297 – cycle AMP should be cyclic AMP

Page 8, line 329 – the word ‘to’ is missing between the words damage and cell.

Page 11, section 4.1 - please address challenges, if any, to achieve receptor subtype selectivity for this class of molecules.

Author Response

Dear Reviewer #1, please accept our apologies for making that mistake. Below you can find point-to-point reply.

Reviewer #1.

This is a short review of the complex field of adenosine receptors, in context of GBMs and brain tumors. The authors succinctly summarize the most important findings that would be of interest to a novice in the field. Necessarily, some of the complexity in ARs, its biological actions, and the challenges/limitations of targeting ARs with respect to GBMs is lost in the process, and could be addressed in revisions to make this review more informative and interesting to the reader.

Some of issues that need to be addressed prior to publication are listed below:

  1. Line 18 – extra period within the sentence pear.to

As suggested, the extra period has been removed

  1. Line 42 – adenosine binds with its receptors…missing s at the end of bind

As suggested, we corrected that error.

  1. The description of points (B) to (F) in the text are mixed up:

Line 45 – (B) and (C) are interchanged/mislabeled in the graphical abstract

 As suggested, the graphical abstract description has been modified (Line 43 to 58).

Point C which is written as a pathological state…is unclear, when referred to the image that shows blood vessels and its regulation.

A pathological state was referred to GBM microenvironment, thus graphical abstract description has been modified.

  1. Line 54 – The description of the last sub-point (D) in the text should be for point (E) in the image.

As suggested, we revised the figure’s caption.

  1. Line 56 - The description of the last sub-point (E) in the text should be for point (F) in the image.

As suggested, we revised the figure’s caption.

  1. Line 100 – Referring to the image creation software as a separate point (B) is confusing because the image contains (A) and (B) which are referred to differently in the preceding text. The same issue also exists in the graphical abstract with point (F). Figures 2 and 3 do not have a separate point referring to the software used.

Based on the reviewer’s comment, we referred to the image creation software at the beginning of the caption in order to avoid misunderstanding.

  1. Page 4, lines 149-166 – As written, this para does a cursory job of introducing adenosine, its complex biological roles, natural origins, differences of its activities compared to ATP/AMP, and receptor subtypes. It would be useful if the expression patterns of different receptors in the body, specially the brain/CNS is discussed. This paragraph should be combined with section 3 (page 6, line 247).

Based on the reviewer’s comment, suggested changes were made.

  1. Page 5 – section 2, lines 167-205: the discussion about GBM pathophysiology should be combined with section 1 (introduction) and should come before discussing adenosine and its receptors.

Thanks to the reviewer for this suggestion.

  1. Page 5 – Section 2.2, lines 221-246 (page 6) should be combined with the discussion of current treatment approaches in the introduction section on page 4.

Thank the reviewer for this suggestion. We provided to combine mentioned section with the discussion

  1. Page 6, line 247 – Section numbering should be 3, not 2.

We apologize, we corrected the numbering

  1. Page 8, line 297 – cycle AMP should be cyclic AMP

As suggested, we revised cycle AMP in cyclic AMP

  1. Page 8, line 329 – the word ‘to’ is missing between the words damage and cell.

As suggested, we provided to revise that word.

  1. Page 11, section 4.1 - please address challenges, if any, to achieve receptor subtype selectivity for this class of molecules.

As suggested, we previous section 4.1 has been changed in section 5.

Reviewer 2 Report

Bova et al discussed introduced brain tumor especially Glioblastoma Multiforme, discussed signaling mediated by adenosine, and reviewed using of adenosine signaling for cancer treatment. Overall, the manuscript is well-written. However, there are many sentences which are very long. This makes it hard to read sometimes. For example, line 277 - 281 is a very long sentence.

The second issue of this manuscript is that the subtitles are not right. For example, there is “2.2” but not “2.1” in the manuscript. While using “2” for “Glioblastoma multiforme pathophysiological features”, the authors also used “2” for “Adenosine and Adenosine Receptors (ARs)” and “the role of adenosine in glioblastoma multiforme”. Moreover, I think the session which talked about “Adenosine receptor antagonists” can just be subtitle under “5”. Before publishing, the subtitles definitely need to be organized well.

Third issue is the resolution of the figures. Due to the low resolution, the figures can’t be zoomed it.

Line 415-416 is a separated paragraph. But there is “in this context” description. Being a separated paragraph, no one would know what the “this” refers to.

The sentence in line 26-29 has two verbs?

Line 227: “with” but not “whit”.

In terms of the content discussed in this manuscript, it is ok.  

Author Response

Thank you for the kind reply. In the main text, you can find all of the revisions highlighted in yellow when referring to the Reviewer #2 observations and below the relative answers point by point. Please see the attachment.

  1. Bova et al discussed introduced brain tumor especially Glioblastoma Multiforme, discussed signaling mediated by adenosine, and reviewed using of adenosine signaling for cancer treatment. Overall, the manuscript is well-written. However, there are many sentences which are very long. This makes it hard to read sometimes. For example, line 277 - 281 is a very long sentence. 

Thank to the reviewer for this comment, we have shortened that sentence period.

  1. The second issue of this manuscript is that the subtitles are not right. For example, there is “2.2” but not “2.1” in the manuscript. While using “2” for “Glioblastoma multiforme pathophysiological features”, the authors also used “2” for “Adenosine and Adenosine Receptors (ARs)” and “the role of adenosine in glioblastoma multiforme”. Moreover, I think the session which talked about “Adenosine receptor antagonists” can just be subtitle under “5”. Before publishing, the subtitles definitely need to be organized well.

As suggested, we provided to reorganize subtitles along the manuscript.

Third issue is the resolution of the figures. Due to the low resolution, the figures can’t be zoomed it.

As suggested, we provided to increase the resolution of all images.

  1. Line 415-416 is a separated paragraph. But there is “in this context” description. Being a separated paragraph, no one would know what the “this” refers to.

“In this context” is referred to GBM context.

  1. The sentence in line 26-29 has two verbs?

We have rephrased the correct sentence.

  1. Line 227: “with” but not “whit”.

Thanks.

  1. In terms of the content discussed in this manuscript, it is ok.  

Thanks.

Reviewer 3 Report

The paper is well written and the  content appropriate.

It can be published with just a few modifications.

On page 5, line 159, we should read: "subsequentially xanthine derivatives were developed......"

-on page 5, a figure showing  the chemical structure of the most relevant A2A antagonists.

On page 11, line 455, check reference 158

Author Response

Thank you for the kind reply. In the main text, you can find all of the revisions highlighted in yellow when referring to the Reviewer #3 observations and below the relative answers point by point. Please see the attachment.

  1. The paper is well written and the  content appropriate.

Thanks to the reviewer for this comment.

  1. It can be published with just a few modifications.
  2. On page 5, line 159, we should read: "subsequentially xanthine derivatives were developed......"

Thanks to the reviewer for this comment. We provided to modify that sentence.

  1. On page 5, a figure showing  the chemical structure of the most relevant A2A antagonists.

Thanks to the reviewer for this suggestion. We added a figure showing the main adenosine receptor antagonists (Figure 4).

  1. On page 11, line 455, check reference 158

As suggested, we checked reference 158 and we provided to replace it with a new one equally significant.

Round 2

Reviewer 1 Report

The authors seem to have addressed most of the issues based on the manuscript. Unfortunately, the cover letter was in a language other than English, and was not reviewed to avoid mistranslations.

The only major critique for the review would be that sections 1 and 2 (introduction, and GBM biology) is rather long, and that the main topic of the review, adenosine receptors, don't appear till page 6 of the 13 page review. While the effort invested in providing a complete picture for the readers is certainly much appreciated, the long prelude may be a distraction from the main topic which is certainly well covered. 

Author Response

Thank you for the kind reply. In the main text, you can find all of the revisions highlighted in yellow when referring to the Reviewer #1 observations and below answers point to point.

The authors seem to have addressed most of the issues based on the manuscript. Unfortunately, the cover letter was in a language other than English, and was not reviewed to avoid mistranslations.

The only major critique for the review would be that sections 1 and 2 (introduction, and GBM biology) is rather long, and that the main topic of the review, adenosine receptors, don't appear till page 6 of the 13 page review. While the effort invested in providing a complete picture for the readers is certainly much appreciated, the long prelude may be a distraction from the main topic which is certainly well covered.

  • We are very sorry for making that mistake. We sent the correct cover letter to the Editor, hoping that You can find there our replies.
  • As suggested, we revised Sections 1 (introduction) and 2 (pathophysiology of GBM), by unifying them and making the main text homogeneous and not too long for the readers. Regarding section 2 (physiopathological characteristics of GBM), considering that most of previous revisions were made exactly in Section 1 and 2, we considered to keep Section 1 as well. Moreover, regarding the main topic (adenosine receptors), notions about the role of adenosine and its receptors in the context of GBM have been added in the Introduction Section avoiding the readers to be distracted from the focus of the article.
